# Live Vaccination with Blood-Stage *Plasmodium yoelii* 17XNL Prevents the Development of Experimental Cerebral Malaria

**DOI:** 10.3390/vaccines10050762

**Published:** 2022-05-11

**Authors:** Takashi Imai, Ha Ngo-Thanh, Kazutomo Suzue, Aoi Shimo, Akihiro Nakamura, Yutaka Horiuchi, Hajime Hisaeda, Takashi Murakami

**Affiliations:** 1Department of Infectious Diseases and Host Defense, Graduate School of Medicine, Gunma University, Maebashi 371-8511, Japan; dr.ngothanhha@gmail.com (H.N.-T.); suzue@gunma-u.ac.jp (K.S.); 2Department of Microbiology, Saitama Medical University, Moroyama-machi, Iruma-gun, Saitama 350-0495, Japan; b21024@student.saitama-med.ac.jp (A.S.); aki_n@saitama-med.ac.jp (A.N.); horiuchi@saitama-med.ac.jp (Y.H.); takmu@saitama-med.ac.jp (T.M.); 3National Hospital for Tropical Disease, 78 Giai Phong, Dong Da, Hanoi 10000, Vietnam; 4Department of Parasitology, National Institute of Infectious Diseases, Tokyo 162-0052, Japan; hisa@niid.go.jp

**Keywords:** malaria, experimental cerebral malaria, live vaccination, inflammatory response, blood–brain barrier

## Abstract

In our work, we aim to develop a malaria vaccine with cross-strain (-species) protection. C57BL/6 mice infected with the *P. berghei* ANKA strain (PbA) develop experimental cerebral malaria (ECM). In contrast, ECM development is inhibited in infected mice depleted of T cells. The clinical applications of immune-cell depletion are limited due to the benefits of host defense against infectious diseases. Therefore, in the present study we attempted to develop a new method for preventing ECM without immune cell depletion. We demonstrated that mice inoculated with a heterologous live-vaccine of *P. yoelii* 17XNL were able to prevent both ECM and lung pathology and survived longer than control mice when challenged with PbA. Live vaccination protected blood–organ barriers from PbA infection. Meanwhile, live vaccination conferred sterile protection against homologous challenge with the *P. yoelii* 17XL virulent strain for the long-term. Analysis of the immune response induced by live vaccination showed that cross-reactive antibodies against PbA antigens were generated. IL-10, which has an immunosuppressive effect, was strongly induced in mice challenged with PbA, unlike the pro-inflammatory cytokine IFNγ. These results suggest that the protective effect of heterologous live vaccination against ECM development results from IL-10-mediated host protection.

## 1. Introduction

Malaria is a parasitic disease that is transmitted by mosquitoes. According to the World Health Organization (WHO) (https://www.who.int/news-room/fact-sheets/detail/malaria) (accessed on 3 January 2022), there were 238,000,000 estimated malaria cases globally in 2000, and 241,000,000 in 2020. Mass drug administration, bed nets, and indoor residual spraying of insecticides have contributed to malaria treatment or prevention [1,2]. Despite this, according to the WHO, 627,000 people were estimated to have died from malaria in 2020. In addition, there is a concerning increase in the incidence of insecticide-resistant mosquitoes and (multi-) drug-resistant malaria parasites, and there have been troubling interactions between malaria infections and the ongoing COVID-19 pandemic [3,4,5] (https://www.who.int/emergencies/diseases/novel-coronavirus-2019) (accessed on 3 January 2022). 

A vaccine is one of the strategies that could be used to control malaria and to overcome the above-mentioned concerns. The development of malaria vaccines has accelerated in recent times. Several types of malaria vaccines [6], including protein subunit vaccines [7,8,9,10,11], DNA vaccines [12], viral vector or virus-like particle vaccines [13,14], whole parasite vaccines, and genetically/chemically attenuated parasite vaccines [15,16,17] are being developed, and some of these are currently in clinical trials. RTS,S/AS01 malaria vaccine (Mosquirix) [8,9,10] is the frontrunner malaria vaccine, and has been administered to 800,000 children in Ghana, Kenya, and Malawi in an ongoing pilot program since 2019. Currently, the WHO recommends using Mosquirix for malaria vaccines in moderate to high transmission areas in Africa. RTS,S/AS01 targets *Plasmodium falciparum* (Pf) circumsporozoite protein (CSP) to block invasion by spolozoite into hepatocytes, and is therefore referred to as a liver-stage vaccine. Although Mosquirix is an innovative vaccine, the vaccine R21 uses the same technology and has shown better efficacy than Mosquirix [15,18]. However, there is one fundamental weakness to this technology: even if only one parasite escapes from host immunity in the liver stage and is released into the bloodstream, a blood-stage infection that leads to the clinical symptoms of malaria can result. The antigenicity of liver- and blood-stage malaria parasites is different; therefore, liver-stage vaccines need to archive sterile immunity within the liver stage, or they cannot protect against blood-stage infections.

As an alternative approach toward the attenuation of malaria, chemical treatments have been used to kill the malaria parasite [16,19]. This strategy involves artificially injecting sporozoites to pass through the liver and invade red blood cells (RBCs), which simulates a natural infection by mosquito bite, and vaccine recipients then take the anti-malaria drug chloroquine to kill the parasite. This vaccine is called PfSPZ-CVac [16,19], and it leads to immunity against both liver- and blood-stage parasites. Genetically attenuated parasites are proposed for a next-generation vaccine. B9/slarp gene-deficient Pf sporozoites were generated [20]; these parasites invaded hepatocytes, but although they died during liver-stage parasite development, they made enough antigen to induce anti-liver-stage malaria immunity. A clinical study of controlled human malaria infection using the above genetically attenuated parasite, PfSPZ-GA1, was conducted in the Netherlands [17]. Usually, sporozoite(s)-based (whole parasite) vaccines have strain specificity. Therefore, whole parasite vaccine can induce sterile immunity against homologous challenge. Whole parasite vaccine cannot induce sterile immunity against heterolgous challenge [21]. Therefore, a versatile malaria vaccine that is effective against a range of malaria parasite strains and species is preferred in the clinical setting. 

Non-replicating viral vectors serve as platforms for vaccines [22]. Chimpanzee adenovirus 63 and modified vaccinia virus Ankara encoding malaria antigen were utilized for a malaria vaccine. Pf liver-stage antigen 1 or liver-stage-associated protein 2 antigens are more effective for inducing an immune response in mice than CSP or pre-erythrocytic antigen thrombospondin-related adhesion protein.

In an effort to gain fundamental knowledge about new vaccine technologies and prevent disease progression, our group has been studying the protective immunity and pathology of malaria using a murine malaria model [23,24,25,26,27,28]. Since CD4-positive T (CD4T) cells and CD8-positive T (CD8T) cells have immunological memory and provide the principle of the mode of action of the vaccine, it is important to determine their role in malaria. 

The role of the immune system during malaria is complicated. For example, T cells show different responses to *Plasmodium yoelii* (Py) and *Plasmodium berghei* ANKA (PbA: lethal strain) at blood stage in a C57BL/6 (B6) mouse malaria model [23,25,26,28,29,30,31,32,33,34]. T cells (helper- and killer-T cells) are protective in Py, but cause pathology in PbA, leading to the development of lethal experimental cerebral malaria (ECM). The immune system therefore acts as a “double-edged sword” during the T cell-mediated immune response to *Plasmodium*. 

Different strains of *Plasmodium*, Py17XNL (PyNL: non-lethal strain) and Py17XL (PyL: lethal strain), and different species of PbA, have different characteristics. PyNL parasites prefer to invade young RBCs (reticulocytes), and are also able to infect the RBC precursors, erythroblasts. Erythroblasts express MHC class I molecules that are recognized by CD8T cells [24], and parasitized erythroblasts are then eliminated by CD8T cells and macrophages. Reticulocytes express some MHC class I molecules that are not recognized by CD8T cells [26]. In contrast, PyL infects both reticulocytes and mature RBCs, while PbA prefers to infect reticulocytes [35], but its infectivity to erythroblasts remains unknown. PyNL has similar features to *P. vivax* (Pv), which prefers to infect reticulocytes. Since human reticulocytes also express MHC class I molecules, CD8T cells can recognize and kill parasitized RBCs [36]. Both CD4T and CD8T cells are involved in ECM. Indeed, experimental depletion of CD4T or CD8T cells by antibodies prolonged the survival of C57BL/6 mice infected with PbA [25,29,30,33,34], and prevented the development of ECM. The role of these immune cells therefore urgently needs elucidation for each different *Plasmodium* species, so that immune cells can be regulated to destroy only the parasite and not the host tissue. Therapeutic strategies using cell-depletion antibodies have been applied in various clinical settings. For example, the depletion antibody against CD20 (a B cell surface antigen), named rituximab [37], has been utilized for treating patients with B-cell lymphoma [38]. Monoclonal antibodies could be very effective for treating malaria immunopathology, but the depletion of target immune cells during infection also changes the immune conditions, leading to considerable problems.

In the present study, we aimed to develop a better vaccine with cross-strain or cross-species protection, in order to establish a strategy for preventing ECM without using T-cell-depletion antibody treatment. Such a strategy would have significant potential for eliminating the global burden of malaria, and in particular for lessening the impact of this disease in the malaria-endemic regions.

## 2. Materials and Methods

### 2.1. Mice

Female C57BL/6 mice aged 6–12 weeks were purchased from SLC (Shizuoka, Japan). Mice were adapted for one week upon arrival and were maintained at a specific pathogen-free animal facility in the Department of Infectious Diseases and Host Defense, Graduate School of Medicine, Gunma University. Mice were bred in an environment of 25 °C under a 12-hour light/12-hour dark cycle with free access to food and water. All animal experiments were reviewed by the Committee for the Ethics of Animal Experiments in the Faculty of Medicine, and were conducted under the control of the Guidelines for Animal Experiments in the Faculty of Medicine, Gunma University, and the law (No. 105) and notification (No. 6) of the Government.

### 2.2. Malaria Parasites and Blood-Stage Infection

Rodent malaria parasites (blood-stage) of PyNL and PyL strains and of PbA (uncloned line) were kindly provided by Prof. M. Torii (Ehime University, Matsuyama, Japan). PyNL and PyL are clonal lines originating from the Middlesex Hospital Medical School, University of London, 1984. PyNL has similarity with the human malaria parasite *P. vivax*, as described in the introduction. *P. vivax* is adapted to humans and has lower mortality for infants in comparison with Pf [39]. Therefore, we used PyNL as a vivax malaria model. Pf induces two different features, namely severe anemia, which resembles PyL, and cerebral malaria, which resembles PbA infection in C57BL/6 mice. Therefore, we used both PyL and PbA as a falciparum malaria model. The parasites were stored in liquid nitrogen. Parasitized red blood cells (pRBCs) were prepared in donor WT mice injected with parasite stock solution intraperitoneally (i.p.). The mice were infected by i.p. injection with 25,000 pRBCs (PyNL) or 50,000 pRBCs (PyL or PbA) from donor mice, suspended in 0.5 mL of RPMI1640 (Sigma-Aldrich, St. Louis, MO, USA). Giemsa staining of peripheral blood smears was used to determine parasitemia. At least 1000 RBCs (or parasitized RBCs) were counted for each sample.

### 2.3. Live Vaccination and Challenge Infection

Blood-stage PyNL (25,000 pRBCs) was used for live vaccination. At forty, 100, or 200 days (indicated in each figure) after the live vaccination, the mice were challenged with 50,000 pRBCs (PyL or PbA), and parasitemia and survival were monitored.

### 2.4. Histopathology

The brains and lungs of naive mice, control (Ctrl) PbA mice infected after day 8, and vaccinated mice challenged with PbA after day 8 were collected after perfusion with PBS (20 mL) followed by paraformaldehyde (4%, 20 mL; FUJIFILM Wako Chemicals Corporation, Osaka, Japan). Collected tissue was placed in 4% paraformaldehyde for 48 h, and then embedded in paraffin using a conventional method. Sections of 2.5 μm thickness were cut using a microtome, mounted on glass slides, and stained with hematoxylin and eosin (H&E). Images were captured using a BZ-710 microscope (Keyence, Osaka, Japan). Data were analyzed using BZ-II software (Keyence).

### 2.5. Evaluation of Blood–Tissue Barrier

To evaluate the integrity of the blood–brain barrier (BBB), blood–cerebrospinal fluid barrier (BCSFB), blood–air barrier (BAB), and alveolar–capillary barrier (ACB), mice were injected with 0.2 mL of 1% Evans Blue solution in PBS (EB: Sigma). Animals were sacrificed after 1 h, and perfused intracardially with 20 mL of PBS. The integrity of the BCSFB was evaluated according to our previous study [29]. Briefly, the skin and muscle were removed from the cisterna magna, where the cerebrospinal fluid (CSF) is visible through the dura mater (Figure 4B). Digital images were obtained, and the blue color of the EB was analyzed using Image J (http://rsbweb.nih.gov/ij/) (accessed on 10 January 2021). Next, the brain and lungs were collected, blotted dry, and soaked in 2 mL of formamide for 48 h at 37 °C, followed by measurement of the absorbance of EB at an optical density (OD) of 630 nm. A standard curve was obtained from serial dilutions of EB solution. 

### 2.6. Antibody ELISA

IgG antibodies against PbA-soluble antigen were assessed by ELISA. Briefly, 96-well ELISA plates (Nunc) were coated with 1 μg/mL in PBS (50 μL/well) overnight at 4 °C. Plates were washed three times with washing buffer (0.05% PBS-Tween-20) and then incubated with blocking buffer (0.5% bovine serum albumin in washing buffer, 200 μL) for 2 h at room temperature (23 °C). Each serum sample was serially diluted (100-, 1000-, and 10,000-fold) with blocking buffer, added to each well (50 μL), and incubated for 2 h at room temperature. After three washes, goat anti-mouse IgG conjugated to horseradish peroxidase (Zymed, South San Francisco, CA, USA; secondary antibodies diluted 1:1000 in blocking buffer) were added (50 μL) to each well for 30 min at room temperature. Alkaline phosphatase substrate (TMB solution; BioLegend, San Diego, CA, USA) was added to induce the enzymatic reaction, and a stop solution (H_2_SO_4_) was added to end the reaction. The absorbance was measured at an OD of 450 nm using a microplate reader.

### 2.7. Flow Cytometry

All antibodies were purchased from BD Biosciences (Franklin Lakes, NJ, USA), eBioscience (San Diego, CA, USA), or BioLegend (San Diego, CA, USA). PE-Cy7 conjugated anti-CD4 (clone: GK1.5), PerCP-conjugated anti-CD8a (clone: 53–6.7), PE-conjugated anti-CD3, APC-conjugated anti-interferon (IFN)γ (clone: XMG1.2), APC-Cy7-conjugated anti-interleukin (IL)-10 (clone: JES5-16E3) antibodies were used to detect cell status, and purified anti-CD16/32 (clone: 2.4G2) antibodies were used for Fc-blocking. Zombie Green (BioLegend) was used for dead cell staining. Single cell suspensions from the spleen were treated with ACK lysis buffer to lyse RBCs and FC-blocked by anti-CD16/32. No in vitro cell stimulation, such as calcium ionophore or PMA, was used, as the intention was to determine spontaneous activation by malaria parasite infection. Fluorochrome-labeled antibodies for surface staining were mixed on ice for 30 min and washed once with FACS buffer, and dead cells were stained with Zombie Green followed by intracellular cytokine staining using conventional methods. Isotype-matched control antibodies were used to evaluate the specific staining. Cells were analyzed using a FACSVerse or FACSAria II flow cytometer (Becton Dickinson, San Jose, CA, USA), and data were analyzed using FlowJo software (Treestar, Ashland, OR, USA). The gating strategy is summarized in Appendix A.

### 2.8. Statistical Analysis

Data are presented as mean ± standard deviation (SD). Values for variables with normal distributions were compared between the experimental group and control group with the two-tailed Student’s *t*-test (Figure 2C, Figure 4A,C, and Figure 5, Figure 6 and Figure 7 ). The Mann–Whitney U-test was used for statistical analysis to compare two sets of data without normal distributions (Figure 4B). A *p*-value of *p* < 0.05 was considered to be statistically significant. Significant differences in survival were tested with a Gehan–Breslow–Wilcoxon test (Figure 2D). GraphPad Prism version 8.0 (GraphPad Software Inc., San Diego, CA, USA) was used in the above analysis.

## 3. Results

### 3.1. Live Vaccination Provided Complete Protection against Homologous Lethal Infection

Infection with blood-stage PyNL was self-limited, and mice infected with PyNL showed peak parasitemia of up to 50% at 2–3 weeks, followed by a complete recovery within a month (Figure 1A). In contrast, the lethal strain PyL killed all mice within 2 weeks along with hyper-parasitemia, which occasionally reached over 90% (Figure 1B). Mice that recovered from PyNL infection were completely protected against lethal infection with PyL without elevated parasitemia (Figure 1C). Therefore, PyNL infection serves as a live vaccination against PyL. The recovered mice were challenged at 100 or 200 days after the primary infection with PyNL, and parasitemia remained suppressed, although one out of ten mice had parasite growth up to 0.65% parasitemia (Figure 1D,E). This suggests that the protection afforded by live vaccination would be long-lasting.

### 3.2. Live Vaccination Prevented the Development of ECM and Lung Pathology Following Heterologous Lethal Challenge

Compared to homologous (cross-strain) challenge infection, heterologous challenge infection has proved to be difficult to address using vaccines [28]. We therefore next evaluated the protective effects of live vaccination against heterologous PbA (cross-species) challenge. C57BL/6 mice infected with PbA suffered from central nervous system symptoms, such as paralysis and convulsions, suggesting the development of ECM. PbA-infected mice died within 2 weeks, even when the parasitemia level was low (Figure 2A,D). Mice that received the live vaccination had the same parasite growth as the control mice at 7 days after PbA infection (Figure 2C). However, vaccinated mice prevented the development of ECM and survived for longer than mice receiving the control PbA infection (Figure 2D). Vaccinated mice (11 mice out of 12) could not suppress PbA parasite growth, and eventually died of hyperparasitemia (Figure 2B, Appendix A).

Histopathological analysis was conducted to confirm the prevention of ECM in the vaccinated mice. When naive mice were sacrificed and perfused with PBS, the brain blood vessels (BVs) did not contain coagulated blood (Figure 3A–C). In contrast, infection of C57BL/6 mice with PbA led to the occlusion of blood vessels in the brain due to sequestration of infected and uninfected red and white blood cells (Figure 3D). Subarachnoid and parenchymal hemorrhages were also observed (Figure 3E,F). These histological lesions, which are typical of ECM, were not found in mice that received live vaccination (Figure 3G–I). Disruption of the integrity of the BBB and BCSFB, another hallmark of ECM development, was further evaluated using an Evans blue dye leakage assay. Since this dye cannot pass through the BBB and BCSFB in healthy mice, naive mice that were intravenously injected with this dye showed no leakage into the cerebral parenchyma and CSF (Figure 4A,B). In contrast, control mice infected with PbA showed leakage of Evans blue dye into the brain parenchyma (Figure 4A), and the CSF (Figure 4B). Mice that received live vaccination with PyNL prior to PbA challenge had no visible leakage of Evans blue dye, indicating that their BBB and BCSFB remained intact (Figure 4A,B). These results suggest that ECM is prevented in live-vaccinated mice due to the protection of the BBB and BCSFB.

Infection with PbA can lead to the development of malaria-associated acute respiratory distress syndrome (MA-ARDS) in addition to ECM [40]. Therefore, we also analyzed the lung pathology in the vaccinated mice. The airways from the bronchus to the alveoli are stringently separated by the BAB or ACB and are usually clear in naive mice (Figure 3J). Mice infected with PbA showed alveolar edema and infiltration, with thickened alveolar walls and occlusion of blood vessels with thrombus-like structures, probably due to sequestration of RBCs and WBCs (Figure 3K). Furthermore, the permeability of the BAB or ACB increased (Figure 4C). No obvious pathological findings were observed in mice vaccinated with PyNL prior to PbA challenge infection (s L and C), indicating that live vaccination prevents lung pathology caused by malaria infection.

### 3.3. Immunological Characteristics of Live Vaccination with PyNL

To determine how live vaccination with PyNL prevents ECM, the immune response in mice vaccinated with PyNL was analyzed. We first tested whether PyNL vaccination stimulated the production of cross-reactive antibodies to PbA. Serum samples of peripheral blood from vaccinated mice immediately prior to PbA challenge contained significant amounts of anti-PbA-specific IgG, while those from unvaccinated (control) mice infected with PbA showed only marginal increases in PbA-specific IgG (Figure 5). 

Next, we analyzed the T cell response in the spleen of mice, since these cells are responsible for the immune reaction during malaria infection [41]. Vaccinated or unvaccinated mice were examined both before and 8 days after PbA challenge. Infection of unvaccinated mice with PbA slightly reduced the percentage of CD4T cells in splenic gated WBCs compared to the percentage in naive mice. PyNL-vaccinated mice had more CD4T cells than unvaccinated mice, although the proportion was similar to that of unvaccinated mice. PbA challenge in vaccinated mice did not result in changes in the proportion or number of CD4T cells compared to those in pre-challenge groups (Figure 6A,B). 

Next, in order to examine the inflammatory status of vaccinated mice, we analyzed the number of T cells producing IFNγ, which is a pro-inflammatory cytokine, and those producing IL-10, which is an anti-inflammatory cytokine, using flow cytometry (Appendix A). Almost no CD4T cells produced IFNγ spontaneously in unvaccinated mice prior to PbA challenge (naive mice: 0.42%, 51 × 10^3^ cells, mean value), but a substantial portion of CD4T cells produced IFNγ in response to PbA infection (control PbA: 1.9%, 162 10^3^ cells, mean value) (Figure 6C,D). Vaccinated mice had more CD4T cells producing IFNγ prior to PbA challenge (Vac pre PbA: 6.7%, 974 × 10^3^ cells, mean value) than the unvaccinated group (control PbA: 1.9%, 162 × 10^3^ cells, mean value). In contrast to unvaccinated mice, CD4T cells producing IFNγ decreased in number after PbA challenge (Vac PbA: 3.4%, 576 × 10^3^ cells, mean value) (Figure 6C,D). The proportion and number of IL-10-producing CD4T cells decreased after PbA infection in unvaccinated mice (naive mice: 4.1%, 466 × 10^3^ cells vs. control PbA: 2.9%, 254 × 10^3^ cells, mean value). Similarly, live vaccination increased the proportion of IL-10-producing CD4T cells compared to unvaccinated mice prior to challenge, and challenge infection with PbA remarkably increased the number of IL-10-producing CD4T cells in vaccinated mice (Vac pre PbA: 6.2%, 908 × 10^3^ cells vs. Vac PbA: 29.5%, 5658 × 10^3^ cells, mean value)(approximately 20-fold more than the unvaccinated control) (Figure 6E,F). Therefore, PbA infection in unvaccinated mice resulted in the development of an ECM-stimulated inflammatory IFNγ response. However, PbA challenge stimulated anti-inflammatory IL-10 production in PyNL vaccinated mice refractory to ECM, which was confirmed by calculating the ratio of IFNγ- and IL-10-producing CD4T cell numbers (Figure 6G).

CD8T cells were also analyzed (Figure 7). Control unvaccinated mice infected with PbA, and vaccinated mice that received pre- and post-challenge infection with PbA, showed a reduced proportion and number of CD8T cells compared to naive mice (Figure 7A,B). Infection of unvaccinated mice with PbA moderately reduced the proportion and the number of IL-10-producing CD8T cells (control PbA: 2.2%, 134 × 10^3^ cells, mean value), but remarkably increased the number of IFNγ-producing CD8T cells (control PbA: 2.6%, 160 × 10^3^ cells, mean value) compared to in naive mice (IL-10: 3.9%, 329 10^3^ cells, IFNγ: 0.2% and 18.3 × 10^3^ cells, mean value) (Figure 7C–E). Vaccination with PyNL did not induce the production of these cytokines (Vac Pre PbA IL-10: 0.3%, 8 × 10^3^ cells, IFNγ: 0.11%, 3.5 × 10^3^ cells, mean value). Infection of vaccinated mice with PbA increased the number of IFNγ-producing CD8T cells (Vac PbA: 1.4%, 88 × 10^3^ cells, mean value) to a lesser extent than the infection of unvaccinated control mice (approximately 50% less) (Figure 7C,D). More IL-10-producing CD8T cells were induced in PyNL-vaccinated mice after infection with PbA (Vac PbA: 16.9%, 1052 × 10^3^ cells, mean value) than in the control mice (approximately 8-fold more than in unvaccinated control) (Figure 7E,F). Consequently, vaccination reduced IFNγ and promoted IL-10 in CD8T cells (Figure 7G), which is similar to what was observed in CD4T cells (Figure 6G).

## 4. Discussion

In the present study, we demonstrated that live vaccination with PyNL provided sterile and long-term immunity against infection with a homologous lethal PyL strain, but not against heterologous PbA infection. The PyL strain was derived from PyNL after cryopreservation [42] and is almost antigenically identical to PyNL. This closeness in identity may have resulted in the induction of sterile immunity. Although the vaccination could not protect mice from PbA infection, it prevented the development of ECM and MA-ARDS in live-vaccinated mice.

Our live vaccination failed to induce sterile immunity against heterologous challenge infection, although previous reports showed that attenuated sporozoite vaccine of Pb or Py protected some mice from heterologous challenge [43]. A blood-stage *P. chabaudi* chemically attenuated vaccine conferred protection against Py or *P. vinckei* [44], hence heterologous protection might be inducible in a rodent malaria model. Chimera Pf expressing PvCSP were generated in [45], and would be among the candidates for a next-generation human cross-protective malaria vaccine. 

The pathogenicity of malaria occurs during the erythrocytic cycle of malaria parasites, and two major pathological mechanisms cause symptoms [2,46,47,48]. One of these is parasite growth, or repeated erythrocytic cycles. The overwhelming destruction of RBCs results, for example, in severe anemia. The other is the immunopathology resulting from excessive host immunity against malaria parasites. Combining these two features complicates the pathogenicity of malaria. The high virulence of PyL is mostly caused by parasitic growth, resulting in death from severe anemia with hyperparasitemia. In contrast, ECM observed following infection with PbA is dependent on immunopathology, and develops while parasitemia is quite low [32].

Protection against malaria is achieved by preventing one or both of these pathological mechanisms. Antibodies specific for parasite antigens and phagocytes play a major role in disrupting the erythrocytic cycle. Phagocytes, such as macrophages, engulf parasites and parasitized RBCs, and are required for the final clearance of malaria parasites [23,26,28,49,50]. Many studies have emphasized the role of antibodies in controlling parasitemia, and almost all malaria vaccines aim to induce specific antibodies [51,52,53]. Antibodies block the motility and invasion of merozoites and egress from RBCs to exert cytolysis of merozoites through antibody-dependent complement-mediated cytotoxicity [54], and to enhance phagocytosis [55]. Live vaccination with PyNL induces antibodies that recognize homologous PyL antigens [23], leading to sterile immunity, thus restricting the erythrocytic cycle of PyL. Although PyNL vaccination also induced cross-reactivity against PbA, it might not be enough to disrupt the erythrocytic cycles of PbA. Further investigations are required to determine the level of cross-reactivity of antibodies produced by PyNL live vaccination against other strains. There is a possibility that live vaccination with PyNL induced cross-reactive antibodies against a common antigen in *Plasmodium.*

The development of ECM occurs via multifactorial immunopathology, and several immune components are involved. For instance, genetic loss of CD4T cells [30], depletion of CD8T cells [25,29,34], and neutralization of IFNγ [56] protected C57BL/6 mice from the development of ECM. Cross-presentation of PbA antigens by endothelial cells in the brain capillary to CD8T cells, followed by cytotoxicity, is one of the mechanisms of ECM [32]. Thus, symptoms due to immunopathology may be prevented by regulating the immune response. The immunosuppressive cytokine IL-10 (which also stimulates antibody (IgA) production) can prevent immunopathology during malaria [57]. Co-infection with filaria, a nematode, parasitizes the lymphatic system and induces IL-10, which prevents the development of ECM, while IL-10 knockout (KO) mice co-infected with filaria developed ECM [58]. Similarly, co-infection with PbA and the non-lethal strain PbXAT prevented the development of ECM in C57BL/6 mice, but not in IL-10 KO mice [59]. In the present study, we demonstrated that PyNL vaccination prevents ECM, probably by regulating the pro-inflammatory status by shifting the balance between IFNγ and IL-10. However, a limitation of the present study is that we did not use IL-10 KO mice, and the exact suppressive mechanisms remain unknown. Future studies using IL-10 KO would help to elucidate these mechanisms. Although mice vaccinated with PyNL were protected from developing ECM and survived for longer than unvaccinated mice, they died from hyperparasitemia due to the vaccine’s failure to prevent PbA parasite growth. Immunosuppressive drugs, such as dexamethasone, prevented MA-ARDS in a mouse model [60]. It is possible that a combination therapy, with live vaccinations to reduce parasitemia and immunosuppressive drugs to prevent excessive inflammation, may be an effective strategy for treating malaria.

The Pf or PbA schizont (Appendix A) has many adhesion molecules. Knob protein, PfEMP1, attaches to host endothelial molecules such as CD36 [47,61,62]. VAR2, a member of PfEMP1, can bind to placental chondroitin sulfate A [63,64]. The Pf NF54 line lacks knob-associated histidine-rich protein (KAHRP), which is the major Knob protein, and has therefore lost its ability to cytoadhere. Meanwhile, the Pf 3D7 line and 7G8 have functional KAHRP and can bind to CD36, thus showing cytoadherence [61]. Due to this cell-adhesion event, it is possible to observe the sequestration or rosette formation of RBCs and WBCs in the blood vessels of the brain and lungs of control PbA-infected mice (Figure 3D and Figure 4C), but not in the live vaccination PbA-challenge-infection group. Schizonts were not observed from peripheral blood smears in control PbA-infected mice, but were observed on day 22 after PbA challenge in vaccinated mice. This may be due to hyperparasitemia, or to the possibility that anti-PbA antibodies might prevent the interaction between the knob protein on schizonts and the host CD36 on endothelial cells. On the other hand, all blood-stage parasites were observed in PyNL-and PyL-infected unvaccinated mice, due to a lack of cytoadherence events. KAHRP may therefore be a good target for malaria vaccines. A group in Australia has been conducting the first clinical investigation using a “genetically attenuated blood-stage human malaria vaccine”, and they generated a parasite that lost KAHRP in Pf 3D7 by gene engineering [65]. We await the results of this clinical investigation eagerly, as it may represent a new paradigm for malaria vaccines.

A recent report demonstrated that ECM was prevented by BCG (bacille Calmette-Guerin)-vaccine [66]. In BCG-vaccinated mice, CD8T cells did not accumulate in the brain of PbA-infected mice. BCG did not show any effect against parasitemia, and it could not save mouse lives following infection. This phenomenon was similar in our observations of mice following live vaccination. 

The mechanism by which cerebral malaria develops in humans and in rodent models remains controversial. Conventionally, the mechanism of development of human cerebral malaria is examined by post-mortem analysis of the brain, and parasite or parasitized RBC sequestration is thought to be the major cause of the pathology [46], rather than immunopathology. In the present study, sequestration of RBCs in the brain BVs was observed (Figure 3). Another study demonstrated that a single PbA pRBC is sufficient to occlude the blood capillaries [67]. Recent reports have suggested that infiltrating CD8T cells were found in the brains of malaria patients [68,69]. Because immunopathology has been reported in other infectious diseases, such as COVID-19 [70,71], there is more than sufficient evidence that it is time to reconsider the immunopathology of CM [31]. The mechanisms of mouse ECM and human CM might not be as different as previously thought [72].

The advantage of live vaccination is that it does not need boost immunization. Live vaccination with PyNL could confer strong immunity to PyL, but vaccine-administered mice had adverse events, such as anemia, fever, and splenomegaly [24,28]. Therefore, we need further attenuation of the parasite to prevent adverse events.

## 5. Conclusions

Live vaccination with blood-stage PyNL provided homologous cross-strain protection and prevented cross-species ECM and MA-ARDS. 

## Figures and Tables

**Figure 1 vaccines-10-00762-f001:**
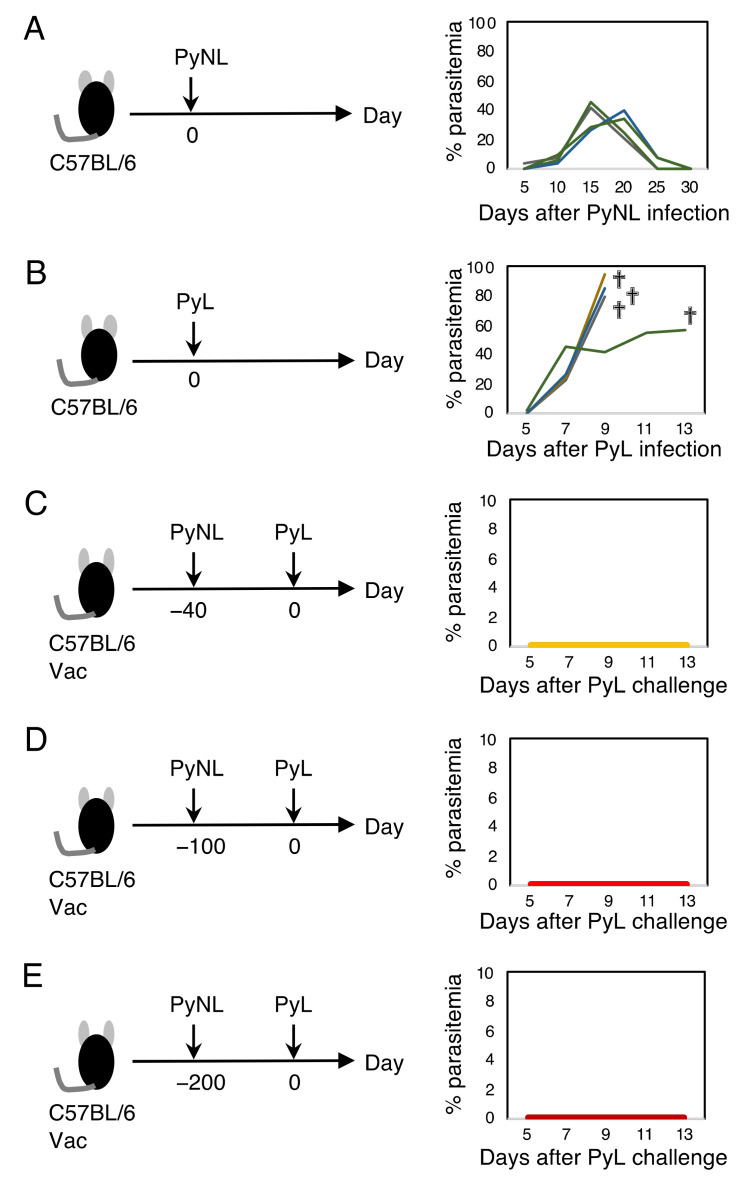
Live vaccination with *Plasmodium yoelii* 17XNL (PyNL) protected mice from the lethal *P. yoelii* 17XL (PyL). (**A**) C57BL/6 mice were infected with PyNL pRBC (left panel) and parasitemia was monitored (right panel). All mice cleared the parasite within one month. Each line shows parasitemia of one mouse (N = 4). A representative dataset is shown from more than 15 different independent experiments. (**B**) C57BL/6 mice were infected with PyL pRBC (left panel) and parasitemia was monitored. Each line shows parasitemia of one mouse (N = 4). “†” indicates death. A representative dataset is shown from more than 10 independent experiments. (C) At forty days after vaccination with PyNL, the mice were challenged with PyL (same as (A) and (B); left panel) and parasitemia was monitored (right panel; N = 57 from pooled independent experiments). There were no parasites detected in PyNL-vaccinated mice after PyL challenge infection (Vac). (D) At one hundred (N = 10) and (E) two-hundred days after Vac (N = 10), the mice were challenged with PyL. Data are pooled from two independent experiments.

**Figure 2 vaccines-10-00762-f002:**
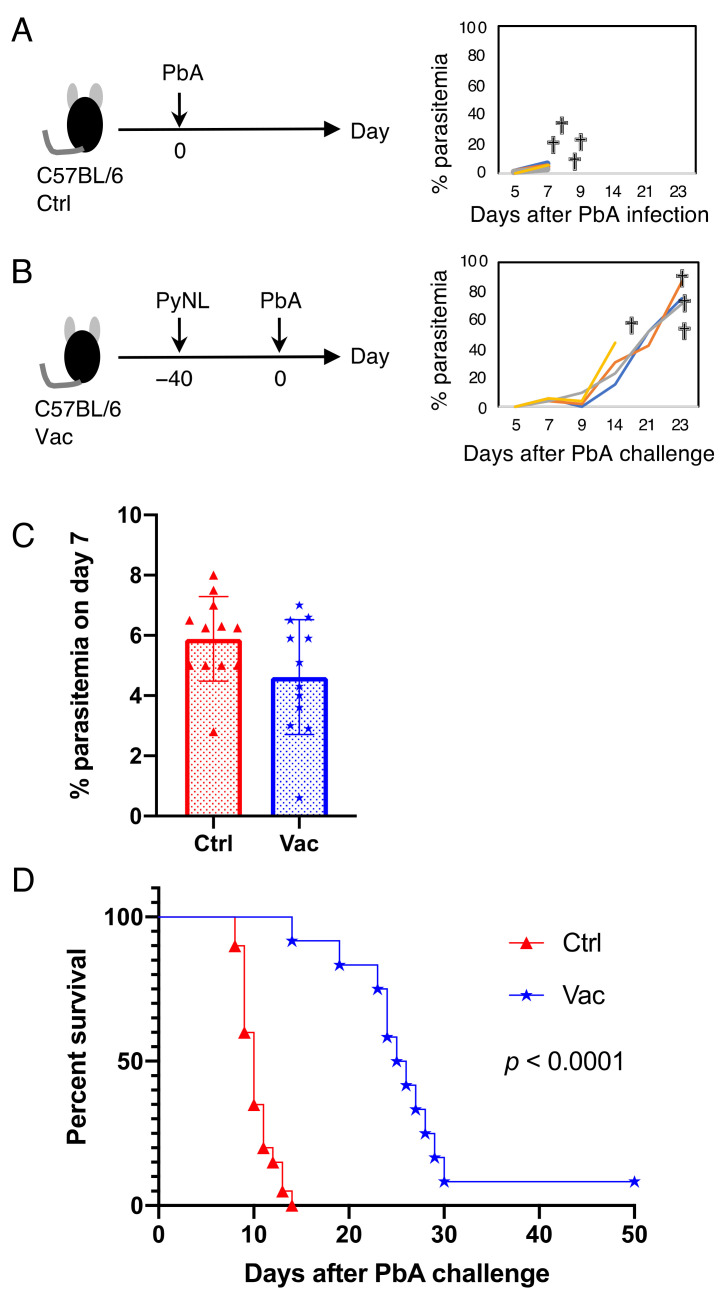
Live vaccination with *Plasmodium yoelii* 17XNL (PyNL) prolonged survival following challenge infection with *P. berghei* ANKA strain (PbA). (**A**) C57BL/6 mice were infected with PbA pRBC (control: Ctrl; left panel) and parasitemia was monitored (right panel). Each line shows parasitemia of one mouse (N = 4). “†” indicates death. A representative dataset is shown from more than 10 independent experiments (N = 4). (**B**) Mice vaccinated with PyNL (Vac) were challenged with PbA (left panel) and parasitemia was monitored (right panel; N = 4). A representative dataset is shown from 3 independent experiments. (**C**) The percentage of parasitemia on day 7 in Ctrl (Red; N = 12) and Vac mice infected with PbA (Blue; N = 12). Each symbol represents one mouse. Data represent the mean ± standard deviation (SD). (**D**) Survival curve of Ctrl (Red; N = 20) and Vac mice infected with PbA (Blue; N = 12). Data were pooled from 3 independent experiments. *p* values were determined by Gehan–Breslow–Wilcoxon test. One mouse out of 12 cleared PbA infection, with peak parasitemia at 0.6% on day 7.

**Figure 3 vaccines-10-00762-f003:**
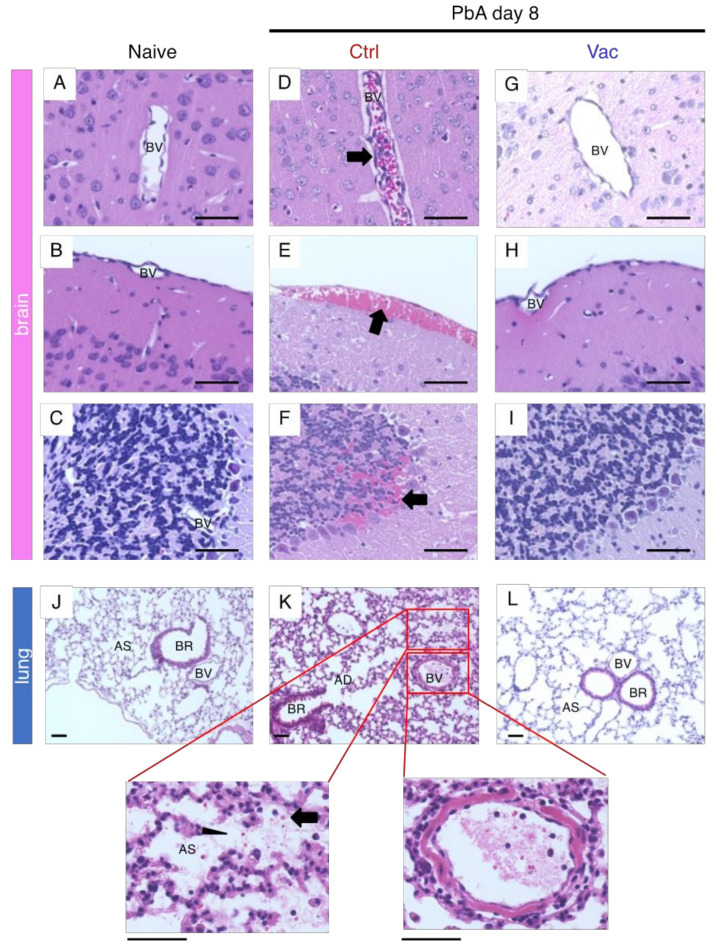
Live vaccination (Vac) with *Plasmodium yoelii* 17XNL protected the mice from the deadly histopathology caused by *P. berghei* ANKA (PbA). C57BL/6 mice (naive mice: N = 6, control (Ctrl) mice infected with PbA on day 8: N = 6, Vac mice infected with PbA on day 8: N = 6) were sacrificed. Representative images of sections from brains and lungs stained with hematoxylin and eosin are shown. (**A**–**C**) Brains of naive mice. (**D**–**F**) Brains of Ctrl PbA infected mice, which developed experimental cerebral malaria. Arrow indicates RBCs and WBCs sequestrated in (**D**) blood vessel (BV), (**E**) subarachnoid hemorrhage, (**F**) hemorrhage found in the cerebellum. (**G**–**I**) Brains of Vac mice infected with PbA on day 8. Scale bar: 50 µm. (**J**,**K,L**) Mouse lungs. (**J**) Naive mouse lung. (**K**) Ctrl mouse lung infected with PbA, arrow indicates RBC found in the alveolar sac (AS). Arrow indicates pulmonary edema in the AS. (**K**) Sequestration of RBCs and WBCs in the artery. The areas surrounded by red lines are enlarged. (**L**) Vac mouse lung infected with PbA. Scale bar: 50 µm. BV (blood vessel); BR (bronchi); AD (alveolar duct).

**Figure 4 vaccines-10-00762-f004:**
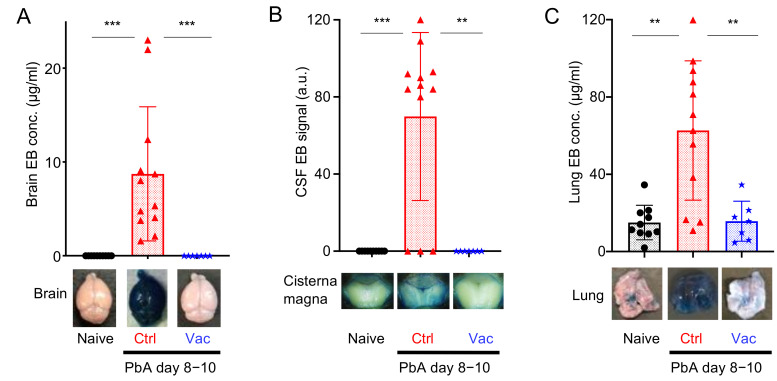
*Plasmodium yoelii* 17XNL live vaccination (Vac) protected the blood–organ barrier from *P. berghei* ANKA strain (PbA) infection. (**A**) Brain Evans blue (EB) concentration indicating the integrity of the blood–brain barrier. (**B**) Cerebrospinal fluid (CSF) EB signal indicating the integrity of the blood–cerebrospinal fluid barrier. (**C**) Lung EB concentration indicating the integrity of the blood–air barrier (alveolar–capillary barrier). Each symbol represents one mouse. Representative photographs of the EB-stained and perfused brain, cisterna magna, and lung from naive (N = 10; black circle), control (Ctrl; N = 12; red triangle), and Vac (N = 7; blue star) mice on days 8–10 following challenge infection with PbA are shown at the bottom of each figure. Data were pooled from 3 independent experiments. Data represent the mean ± standard deviation (SD). ** *p* < 0.01; *** *p* < 0.001. *p* values were determined by two-tailed Student’s *t*-test or Mann–Whitney U-test.

**Figure 5 vaccines-10-00762-f005:**
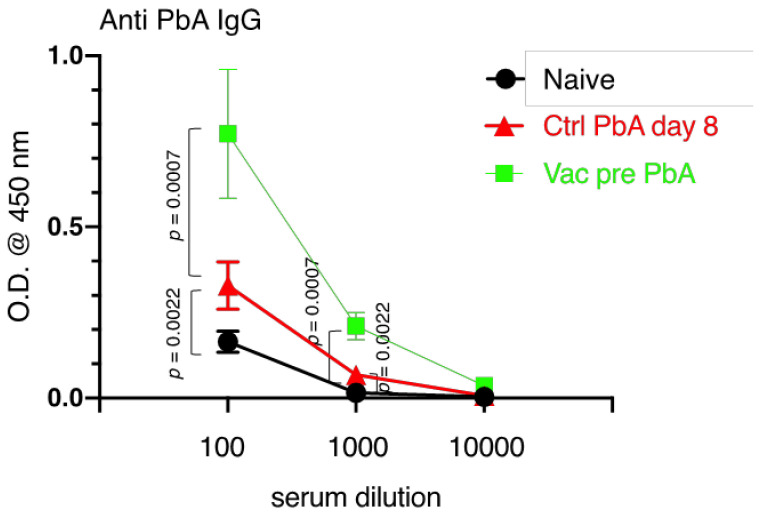
*Plasmodium yoelii* 17XNL live vaccination (Vac) induced cross-reactive antibody production against *P. berghei* ANKA (PbA). Serum samples collected from naive mice (N = 6; black circle), control (Ctrl) mice infected with PbA on day 8 (N = 6; red triangle), Vac mice pre-infected with PbA at 40 days after vaccination with *Plasmodium yoelii* 17XNL (N = 8; green square) were diluted 100-, 1000-, 10,000-fold, and PbA antigen-specific IgG was detected by ELISA. The optical density (O.D.) at 450 nm is shown. Data represent the mean ± standard deviation (SD). *p* values were determined by two-tailed Student’s *t*-test. A representative dataset is shown from 3 independent experiments.

**Figure 6 vaccines-10-00762-f006:**
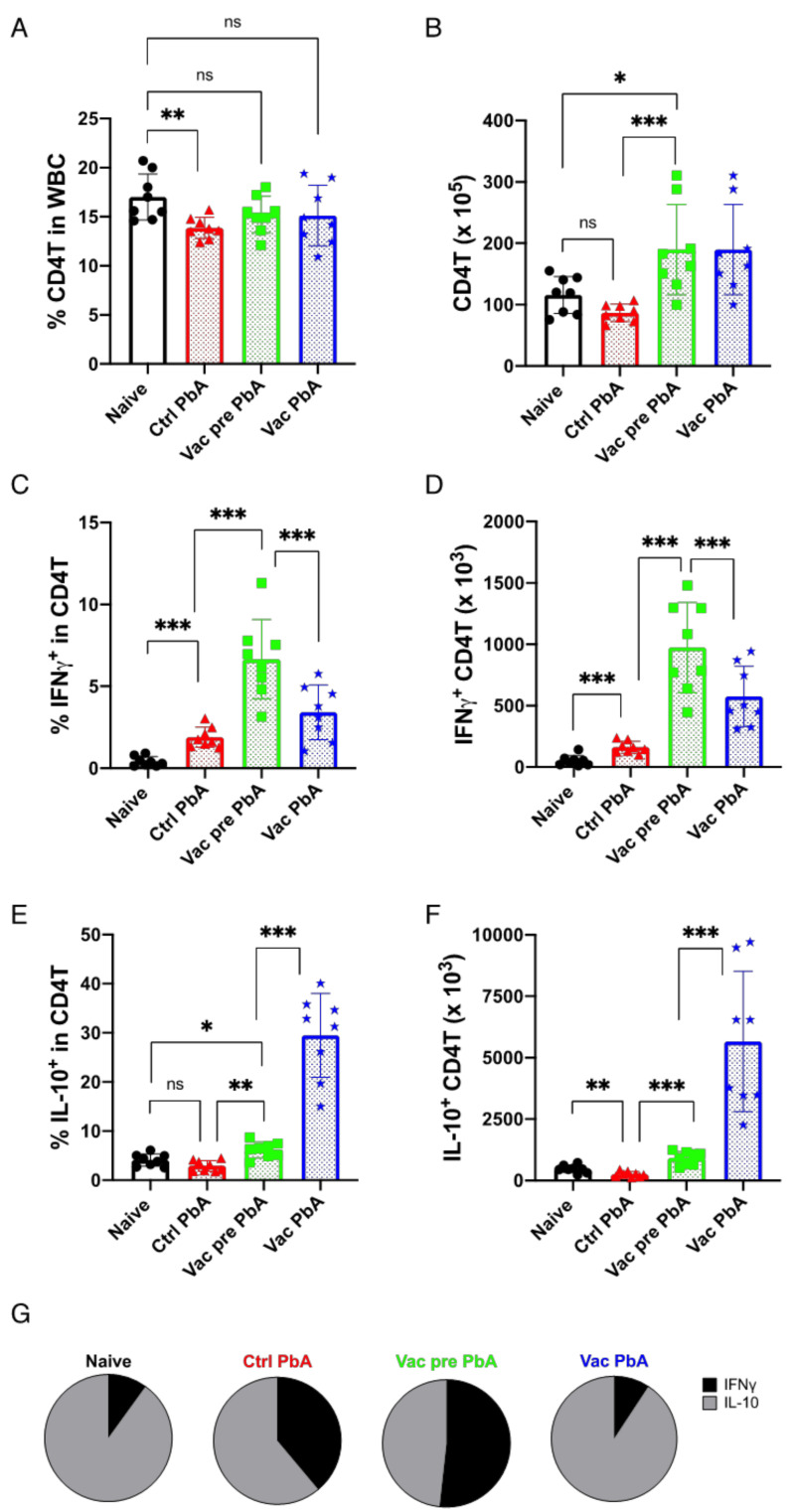
*Plasmodium yoelii* 17XNL live vaccination (Vac) tuned the balance between the pro- and anti-inflammatory response in splenic CD4^+^ T cells against *P. berghei* ANKA (PbA) challenge infection. Splenocytes from naive mice (N = 8; black circle), control (Ctrl) PbA mice infected on day 8 (N = 8; red triangle), Vac mice pre-infection and 40 days after vaccination with *Plasmodium yoelii* 17XNL (N = 8; green square), and Vac mice infected with PbA on day 8 (N = 8; blue star) were obtained to analyze intra-cellular cytokine production. (**A**) Proportion of CD4^+^ CD3^+^ (CD4T) cells in WBCs. (**B**) Absolute number of CD4T cells in the spleen. (**C**) Proportion of IFNγ producing cells in CD4T cells. (**D**) Absolute number of IFNγ-producing CD4T cells in the spleen. (**E**) Proportion of IL-10-producing cells in CD4T cells. (**F**) Absolute number of IL-10 producing CD4T cells. (**G**) Ratio of IFNγ-producing CD4T cells (black) to IL-10-producing CD4T cells (gray). Data are pooled from 2 independent experiments. * *p* < 0.5; ** *p* < 0.01; *** *p* < 0.001; ns: not significant. Each symbol represents one mouse. Data represent the mean ± standard deviation (SD). *p* values were determined by two-tailed Student’s *t*-test.

**Figure 7 vaccines-10-00762-f007:**
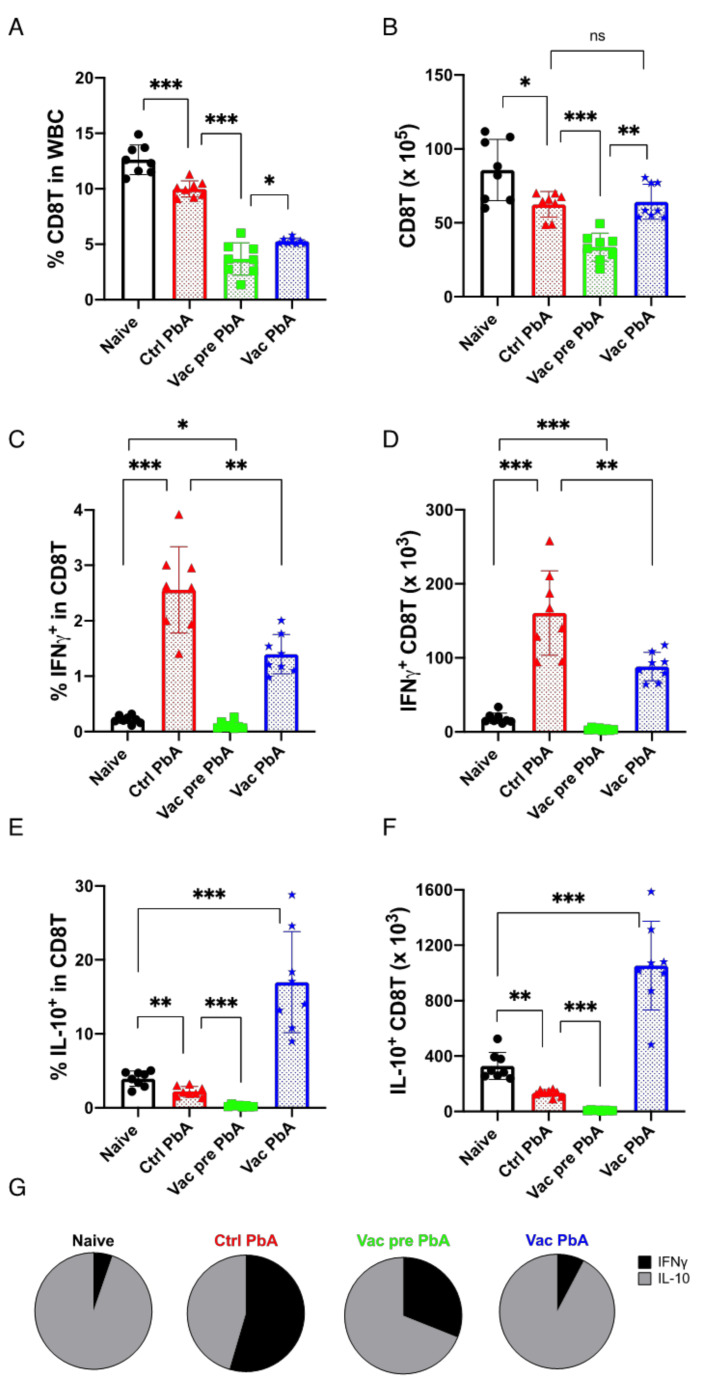
Live vaccination (Vac) tuned the balance between the pro- and anti-inflammatory response in splenic CD8^+^ T cells against *P. berghei* ANKA (PbA) challenge infection. Splenocytes from naive mice (N = 8; black circle), control (Ctrl) PbA mice infected on day 8 (N = 8; red triangle), Vac mice pre-infection at 40 days after vaccination with *Plasmodium yoelii* 17XNL (N = 8; green square), and Vac mice infected with PbA on day 8 (N = 8; blue star) were obtained to analyze intra-cellular cytokine production. (**A**) Proportion of CD8^+^ CD3^+^ (CD4T) cell in WBCs. (**B**) Absolute number of CD8T cells in the spleen. (**C**) Proportion of IFNγ-producing cells in CD8T cells. (**D**) Absolute number of IFNγ-producing CD8T cells in the spleen. (**E**) Proportion of IL-10-producing cells in CD8T cells. (**F**) Absolute number of IL-10-producing CD8T cells. (**G**) The ratio of IFNγ-producing CD8T cells (black) to IL-10-producing CD8T cells (gray). Data are pooled from 2 independent experiments. *: *p* < 0.5; **: *p* < 0.01; ***: *p* < 0.001; ns: not significant. Each symbol represents one mouse. Data represent the mean ± standard deviation (SD). *p* values were determined by two-tailed Student’s *t*-test.

## Data Availability

Data is contained within the article.

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
