# Peer review of "Live Vaccination with Blood-Stage *Plasmodium yoelii* 17XNL Prevents the Development of Experimental Cerebral Malaria"

_vaccines, 2022, doi:10.3390/vaccines10050762_

Round 1

Reviewer 1 Report

This manuscript describes the observation that immunity of self cured P.y. 17XLN infected mice inhibited the development of experimental cerebral malaria (ECM) caused by P.b. ANKA (PbA) infection, and that a possible mechanism of ECM development may related to IL-10-mediated host protection. In addition, authors argued species cross reactive antibody induced by  P.y. 17XLN infection. The authors, in the Discussion section, did recognize an important limitation—that they were not able to elucidate the induction mechanism of IL-10+ CD8T cell by PbA infection in post P.y. 17XLN infected mice. I would add also that cross reactive antibodies to PbA is not really not specific (PbA soluble antigen) but would be common antigen in Plasmodium and not strong proof to conclude cross-protection.  A major criticism/caveat is that the authors tended to overemphasize the observed results towards cross-species protection and concept of live-vaccine that could easily mislead/hype readers at this very early stage of development (e.g. even for the whole sporozoite vaccine with chemoprophylaxis with choloroquine safety aspects together with vaccine efficacy are being resolved in terms of dose, schedule, possible alloantibodies, etc).  The manuscript maybe relevant in journals of immunology field but not in vaccinology field.

Other comments:

  1. Statements in the abstract: “Live-vaccines may be applied in humans for the prevention of cerebral malaria using heterologous strains.” And in the conclusion: “The mechanisms of mouse ECM and human CM might not (be) as different as previously thought [68].” – are best toned down especially in the absence of further concrete proof.

  1. Introduction:  A brief summary of malaria vaccines would suffice to rather than two paragraphs in the introduction.Also, WHO recommendation for RTS,S/AS01 is only for children in moderate to high transmission areas in Africa (not whole sub-Saharan Africa). Be careful of the description for PfSPZ-CVac—the vaccine strategy is simultaneous treatment with chloroquine

Line 74 in introduction should also be toned down, as authors should also present studies that do not show direct correlation/association that future human development will be based on pre-clinical studies in mice. Further limitation of a generalized sentence is also see in Line 88—so it is best to tone this sentence as it can be misleading.

Line 81- the role of the immune system during malaria “is not completely understood”?

Line 105 on monoclonal antibodies needs a reference

Line 113 – what does Global South mean?

Figure legends (“Dagger”) indicates death, is best expressed using the symbol used by the authors, eg “†”, indicates death

  1. Line 222, rewording—heterologous challenge infection in itself is not difficult to address—but rather strain/allele specific vaccine efficacy.

  1. Fig. 3:  J-K should be J,K. Some use of “however” and “in contrast” in the abstract is a bit misleading.

Author Response

Reviewer1

Comments and Suggestions for Authors

This manuscript describes the observation that immunity of self cured P.y. 17XLN infected mice inhibited the development of experimental cerebral malaria (ECM) caused by P.b. ANKA (PbA) infection, and that a possible mechanism of ECM development may related to IL-10-mediated host protection. In addition, authors argued species cross reactive antibody induced by  P.y. 17XLN infection. The authors, in the Discussion section, did recognize an important limitation—that they were not able to elucidate the induction mechanism of IL-10+ CD8T cell by PbA infection in post P.y. 17XLN infected mice. I would add also that cross reactive antibodies to PbA is not really not specific (PbA soluble antigen) but would be common antigen in Plasmodium and not strong proof to conclude cross-protection.  A major criticism/caveat is that the authors tended to overemphasize the observed results towards cross-species protection and concept of live-vaccine that could easily mislead/hype readers at this very early stage of development (e.g. even for the whole sporozoite vaccine with chemoprophylaxis with choloroquine safety aspects together with vaccine efficacy are being resolved in terms of dose, schedule, possible alloantibodies, etc). 

Response: Thank you very much for your suggestion. We added “There is a possibility that live-vaccination with PyNL induced cross-reactive antibody against common antigen in Plasmodium” line: 424 (446). We agreed with the statement that cross-reactive antibody detected in our live-vaccination is “not strong proof to conclude cross-protection”, because live-vaccination with PyNL did not decrease the PbA parasitemia and immunize mice eventually died with hyper-parasitemia. Accordingly, in terms of cross-protective vaccine efficacy, our vaccine strategy failed. Also, we do not plan to inject rodent parasite into humans for vaccination now. For this reason, we deleted “overemphasized expression” throughout the text.

Important points that we wanted to emphasize are that live-vaccination with PyNL induced sterile protection against lethal strain PyL and prevented the mice from development of ECM caused by cross-species PbA.

The manuscript maybe relevant in journals of immunology field but not in vaccinology field.

Response: According to MDPI Vaccines “About Vaccines” https://www.mdpi.com/journal/vaccines/about, “immune responses to vaccines” is one of the subject areas within the journal’s scope. Therefore, we believe our article fits the journal’s focus.

Other comments:

Statements in the abstract: “Live-vaccines may be applied in humans for the prevention of cerebral malaria using heterologous strains.” And in the conclusion: “The mechanisms of mouse ECM and human CM might not (be) as different as previously thought [68].” – are best toned down especially in the absence of further concrete proof.

Response: Thank you very much for your comment. “Live-vaccines may be applied in humans for the prevention of cerebral malaria using heterologous strains.” is now deleted.

The statement “The mechanisms of mouse ECM and human CM might not (be) as different as previously thought (Craig, A.G. et al.; PLoS Pathog 2012, doi:10.1371/journal.ppat.1002401)” is based on our discussion section “The mechanism by which cerebral malaria develops in humans and in rodent models remains controversial. Conventionally, the mechanism of development of human cerebral malaria is examined by post-mortem analysis of the brain, and parasite or parasitized RBC sequestration is thought to be the major cause of the pathology, rather than immunopathology (Aikawa, M. et al.; Am J Trop Med Hyg 1990, doi:10.4269/ajtmh.1990.43.30.). In the present study, sequestration of RBCs in the brain BVs was observed (Fig. 3). Another study demonstrated that a single PbA pRBC is sufficient to occlude the blood capillaries (Strangward, P. et al.; PLoS Pathog 2017, doi:10.1371/journal.ppat.1006267.). Recent reports have suggested that infiltrating CD8T cells were found in the brains of malaria patient (Barrera, V. et al.; Front Immunol 2019, doi:10.3389/fimmu.2019.01747., Riggle, B.A. et al.; J Clin Invest 2020, doi:10.1172/JCI133474.)”. We are not going to tone down but we moved that into the last part of the discussion section (Line 473, (505) ). As technology evolves day by day, it is necessary to evaluate phenomena from various angles and constantly re-examine old knowledge.

Introduction:  A brief summary of malaria vaccines would suffice to rather than two paragraphs in the introduction. Also, WHO recommendation for RTS,S/AS01 is only for children in moderate to high transmission areas in Africa (not whole sub-Saharan Africa). Be careful of the description for PfSPZ-CVac—the vaccine strategy is simultaneous treatment with chloroquine

Response: Thanks for your comment and information. This journal is not specific for malaria research, so we thought an introduction for malaria would be essential. But we agree with your points, and we have shortened the summary of malaria vaccines in the introduction.

Line 74 in introduction should also be toned down, as authors should also present studies that do not show direct correlation/association that future human development will be based on pre-clinical studies in mice. Further limitation of a generalized sentence is also see in Line 88—so it is best to tone this sentence as it can be misleading.

Response: “It is conceivable that future human vaccine development will be based on pre-clinical studies in mice [21-23]” was deleted. Thanks for raising this point.

Line 81- the role of the immune system during malaria “is not completely understood”?

Response: “The role of the immune system during malaria is complicated” is what we intended (Now revised in line 84 (94) ).We would like to describe the complexity of the immune response against malaria; complex because it eliminates the parasite and sometimes causes immunopathology.

Line 105 on monoclonal antibodies needs a reference

Response: We added a reference about monoclonal antibody rituximab (Line 108 (118) ).

Line 113 – what does Global South mean?

Response: We apologize; it was typo. It now reads “the malaria endemic regions” (Now line 116 (126) ).

Figure legends (“Dagger”) indicates death, is best expressed using the symbol used by the authors, eg “†”, indicates death

Response: We appreciate your suggestion; we modified the figure captions accordingly.

Fig. 3:  J-K should be J,K. Some use of “however” and “in contrast” in the abstract is a bit misleading.

Response: Thanks for highlighting that; we modified the Figure legend accordingly.

We have reviewed the use of “however” and “in contrast” in the abstract and made some modifications. (Lines 13 and 20).

Reviewer 2 Report

Minor points:

line 14: Missing word? -> "in infected mice depleted" of " T cells".

Line 116, M&M: were only female mice used? Please discuss whether the observed effects are also found in male mice.

Results, long-term experiments: how high are the IgG titers and how high is the target-specific T cell frequency in the mice (groups 100 or 200 days after PyNL vacc. mice)? Was this recorded?

Figure 3: Could the results be quantified?

Fig. 5-7: Are there changes in serum IL-10 levels in the different groups?

Author Response

Reviewer 2

We appreciate the time and effort the Editor and Reviewers have dedicated to providing their insightful feedback.

Minor points:

line 14: Missing word? -> "in infected mice depleted" of " T cells".

Response: Thank you for pointing this out. We modified the text accordingly.

Line 116, M&M: were only female mice used? Please discuss whether the observed effects are also found in male mice.

Response: We used only female mice for the current vaccine experiments, so we do not know about differences in immune response between male and female mice. In previous experiments we used male and female mice for live-vaccination experiments, and we confirmed that live-vaccination with PyNL protected both male and female mice from lethal strain PyL (Imai et al., Vaccines 2020).

Results, long-term experiments: how high are the IgG titers and how high is the target-specific T cell frequency in the mice (groups 100 or 200 days after PyNL vacc. mice)? Was this recorded?

Response: We did not conduct the antibody and T cell analysis at 100 days and 200 days after live-vaccination with PyNL. We would like to do such an experiment in the next project. Thank you so much for your suggestion.

Figure 3: Could the results be quantified?

Response: In our opinion, quantification of histology in H and E staining is not suitable in this case, because we usually use 3-6 sections of 2.5 µm thickness per brain, and sometimes we did not observe the abnormal findings even in ECM mice. We believe this finding is due to the limited searching area. Another researcher says disruption of one micro vessel is sufficient to develop ECM. Such a thing is easily missed from our observation.

Cutting and searching sections from the whole brain was impossible for us. Therefore, we utilized Evans blue dye to quantify the pathology in Figure 4, instead of quantitively analyzing histology sections.

Fig. 5-7: Are there changes in serum IL-10 levels in the different groups?

Response: We tried to detect serum IL-10 level by ELISA for many years but we have never succeeded. We used the ELISA kit produced by Biolegend which worked for the detection of IL-10 in spleen lysate (Imai, T.; Suzue, K.; Ngo-Thanh, H.; Ono, S.; Orita, W.; Suzuki, H.; Shimokawa, C.; Olia, A.; Obi, S.; Taniguchi, T., et al. Fluctuations of Spleen Cytokine and Blood Lactate, Importance of Cellular Immunity in Host Defense Against Blood Stage Malaria Plasmodium yoelii. Front Immunol 2019, 10, 2207, doi:10.3389/fimmu.2019.02207.).

Reviewer 3 Report

Overall the anuscript is interesting. It focueses in an area of interest as it is the development of vaccines  efficient against cerebral malaria. There are very few current alternatives in clinical practice. So it is an area worhty to explore. The re are few comments to beaddressed before publication:

  1. I woudl like to see more info regarding the type of vaccine and its characteristics.
  2. Also, which type of admsinitration route do you think is more suitable for this type of vaccine?
  3. I wonder if the administration of the vaccine regarding the infection is suibale, have the immune response being developed suitable?
  4. Coudl the vaccine being useful post infection?
  5. Discussion shoudl be improved compared with other published works. 

Author Response

Reviewer3

Comments and Suggestions for Authors

Overall, the manuscript is interesting. It focuses in an area of interest as it is the development of vaccines  efficient against cerebral malaria. There are very few current alternatives in clinical practice. So it is an area worthy to explore.

Response: We appreciate the time and effort the Editor and Reviewers have dedicated to providing their insightful feedback. Thank you so much for your positive comment.

There are few comments to be addressed before publication:

1. I would like to see more info regarding the type of vaccine and its characteristics.

Response: Several types of malaria vaccines, including protein subunit vaccines, DNA vaccines, viral vector or virus-like particle vaccines, whole parasite vaccines, or genetically/chemically attenuated parasite vaccines, are being developed. Subunit vaccine such as RTS, S/AS01, R21, and whole parasite vaccine such as PfSPZ-CVac were already described in the manuscript. Now we have added information about another malaria vaccine (PfSPZ-GA1), ChAd63, and MAV virus vector vaccine in the introduction (line: 63 (72) ).

2. Also, which type of administration route do you think is more suitable for this type of vaccine?

Response: We used the i.p. route for our live-vaccination to induce infection; i.v. is also good. Malaria parasites infect the RBC, so we need to allow parasites to circulate in the blood stream and make parasites multiplyenough to induce an immune response. We did a pilot study in which mice received oral administration of PyNL and in this case infection did not occur. In that case, when we challenged the mice with PyL there was no protection and mice even took the parasites into the body. That result clearly showed that to allow parasite infection we need to inject the parasite by the i.p. or i.v. route. These comments would disrupt the flow of the article, so we just put the response here.  

3. I wonder if the administration of the vaccine regarding the infection is suibale, have the immune response being developed suitable?

Response: Thanks for your comment. We believe the immune response primed by infection is suitable; in other words, attenuated live-vaccination is controlled infection because we do not use protein, DNA, RNA, or killed pathogens but we use naturally attenuated live-pathogens (parasites) for inducing strong immunity. Live-vaccination’s advantage is that it does not need boost immunization, whereas the reviewer’s concern may be correct, because live-vaccination with PyNL could confer strong immunity to PyL but the vaccine-administrated mice had adverse events such as anemia, fever, and splenomegaly.

This concern was reported in our previous paper (Imai et al.; Sci Rep 2013, Vaccines 2020). We concluded that further attenuation of the parasite is needed to prevent adverse events. We wondered whether further attenuation to weaken the efficacy of vaccination, followed by boost immunization might be needed.

Some of above comments were added in the discussion. (Lines 484-7 (507-510) ).

4. Could the vaccine being useful post infection?

Response: We never tested the therapeutic vaccine effect. We would like to do such an experiment in the next project. Thank you so much for your suggestion.

5. Discussion should be improved compared with other published works. 

Response: We added other published works in discussion.

Reviewer 4 Report

In this manuscript, the authors use as model vaccine, a live attenuated strain of a rodent malaria species (P. yoelii, PyNL) to demonstrate its protective effect against ECM (Experimental Cerebral Malaria) caused by another species (P. berghei). They suggest the knowledge gained with this rodent model could be used to develop such live malaria vaccine for preventing cerebral malaria (CM) in humans. Few comments on the choice of these rodent parasite species, as well as which malaria parasite species could possibly be used in humans, would enhance the article.

A comment on the use of attenuated sporozoites of different Plasmodium species in cross-protection vaccine experiments (e.g., Parasite Immunol. 2007 Nov; 29(11): 559–565), important in the context of this article, would also enhance the article.

Title, as well as description of contents of images, in legend of Figures must be improved. There are many descriptions that would be more appropriate in Materials and Methods.

The authors must revise the entire manuscript for:
. Spell checking:
- P. berghrei (P. berghei), lines 251, 264, 279, Fig 5 line 309, 338, 385, 483
- spolozoites (sporozoite), line 56
- observe (observed) line 403
- presnent (present), line 433
- and possibly others

. Ambiguous or unnecessary sentences such as:
- "...there are different roles of T cells in the blood stage between..." lines 81, 82;
Suggestion: ".For instance, T cells show different responses to Plasmodium yoelii (Py) and Plasmodium berghei ANKA (PbA: lethal strain), at blood stage, in a C57BL/6 (B6) mouse malaria model [refs.]"  

- "Both CD4T and CD8T cells can cause ECM." line 98;
Suggestion: "Both CD4T and CD8T cells are involved in ECM."

- "...probably due to differences between PyNL and PbA,..." line 381. Obs.: Necessary?

- "Each symbol represents one mouse." lines 285, 348, 395. Obs.: Each point in the graphic? 

The following term may be avoided:
- "...Global South."  line 113; Suggestion: "...in malaria endemic regions."

Author Response

Reviewer4

Comments and Suggestions for Authors

In this manuscript, the authors use as model vaccine, a live attenuated strain of a rodent malaria species (P. yoelii, PyNL) to demonstrate its protective effect against ECM (Experimental Cerebral Malaria) caused by another species (P. berghei). They suggest the knowledge gained with this rodent model could be used to develop such live malaria vaccine for preventing cerebral malaria (CM) in humans. Few comments on the choice of these rodent parasite species, as well as which malaria parasite species could possibly be used in humans, would enhance the article.

Response: Thanks for your comment. The reason why we choose PyNL was described in:

Line: 99 (109). ”PyNL has similar features to P. vivax (Pv), which prefers to infect reticulocytes.”

Line 133 (143) : “P. vivax is adapted to humans and has lower mortality in comparison with P. falciparum.”

And Line 135 (145) : “P. falciparum induces two different features namely, severe anemia which resembles PyL, and cerebral malaria which resembles PbA infection in C57BL/6 mice. Therefore, we used both PyL and PbA as a …”.

The following statement in the previous version of the abstract “Live-vaccines may be applied in humans for the prevention of cerebral malaria using heterologous strains.” was criticized by another reviewer. We partially accept their criticism, because our live-vaccination with PyNL did not decrease the PbA parasitemia and immunized mice eventually died with hyper-parasitemia. Accordingly, in terms of cross-protective vaccine efficacy, our vaccine strategy looked like a failure. And also, we do not plan to inject rodent parasites into humans for vaccination now. One important thing is that the malaria parasite has host tropism (e.g. rodent malaria parasite is not able to infect human RBC; human malaria parasite is not able to infect rodent RBC). Thus, a rodent malaria parasite may be usable for immunomodulation to control the human malaria pathology. However, we do not want to create any conflict thus the related expressions were deleted.

A comment on the use of attenuated sporozoites of different Plasmodium species in cross-protection vaccine experiments (e.g., Parasite Immunol. 2007 Nov; 29(11): 559–565), important in the context of this article, would also enhance the article.

Response: Thanks for letting us know of that article; we included it in the discussion. (Line 385 (405) )

Title, as well as description of contents of images, in legend of Figures must be improved. There are many descriptions that would be more appropriate in Materials and Methods.

Response: Since figures and captions appear online separate to the rest of the article, and must make sense when not accompanied by the main text, we prefer not to shorten the captions. Some Titles of Figures and Figure legends were a bit simplified, and also, garbled characters were found inside the image of Figure 4, therefore we replaced it. We hope it’s acceptable.

The authors must revise the entire manuscript for:
. Spell checking:
- P. berghrei (P. berghei), lines 251, 264, 279, Fig 5 line 309, 338, 385, 483
- spolozoites (sporozoite), line 56
- observe (observed) line 403
- presnent (present), line 433
- and possibly others

Response; Thank you very much for pointing this out; we revised them.

Ambiguous or unnecessary sentences such as:
- "...there are different roles of T cells in the blood stage between..." lines 81, 82;
Suggestion: ".For instance, T cells show different responses to Plasmodium yoelii (Py) and Plasmodium berghei ANKA (PbA: lethal strain), at blood stage, in a C57BL/6 (B6) mouse malaria model [refs.]"  

- "Both CD4T and CD8T cells can cause ECM." line 98;
Suggestion: "Both CD4T and CD8T cells are involved in ECM."

Response: We appreciate your suggestions and we revised the text. (Lines 84   (97) and 101 (111) ).

- "...probably due to differences between PyNL and PbA,..." line 381. Obs.: Necessary?

Response: Thanks for your comment. We have deleted that.

- "Each symbol represents one mouse." lines 285, 348, 395. Obs.: Each point in the graphic? 

Response: Thanks for your comment. We prefer to retain “symbol”.

The following term may be avoided:
- "...Global South."  line 113; Suggestion: "...in malaria endemic regions."

Response: Sorry, it was a typographical error. We revised the text as you suggested. (Now line 116 (126) ).